# The roles and signalling pathways of lncMALAT1 in coronary artery disease: A protocol for systematic review of in vivo and in vitro studies

Jia Zheng[1,2], Arimi Fitri Mat Ludin[3], Nor Fadilah Rajab[3], Li Shaolong[4],
Nurul Farhana Jufri[2*]

1 Department of Cardiovascular Surgery, Yan'an Hospital affiliated to Kunming Medical University, Kunming, China, 2 Center for Toxicology and Health Risk Studies, Faculty of Health Sciences, Universiti Kebangsaan Malaysia, Kuala Lumpur, Malaysia, 3 Center for Healthy Aging and Wellness, Faculty of Health Sciences, Universiti Kebangsaan Malaysia, Kuala Lumpur, Malaysia, 4 Department of Cardiology, Yan'an Hospital affiliated to Kunming Medical University, Kunming, China

* nurulfarhana@ukm.edu.my

## Abstract

### Background

Coronary artery disease (CAD) is a major cardiovascular disease that affects global population health. Several studies have indicated the association between high expression level of a non-coding RNA, lncMALAT1 and an increased risk of CAD. In this study, we conducted a protocol of systematic review aims to evaluate the role and mechanism of lncMALAT1 that may contributed to CAD based on animal and in vitro studies. The roles of lncMALAT1 will be elucidated focusing on activating upstream signalling Klotho/FGF23 or regulate the downstream Wnt/β-catenin or extracellular signal-regulated kinase/mitogen-activated protein kinase(ERK/MAPK) and any other pathways with the vascular changes in term of proliferation, migration, lumen formation and apoptosis.

### Methods

A systematic review protocol with a reproducible strategy according to the Preferred Reporting Items for Systematic Review and Meta-analysis Protocols (PRISMA-P) guidelines and Population, Intervention, Comparison Outcome and Study (PICOS) framework were proposed to evaluate the existing literature on the roles and mechanisms of the lncMALAT1. A PRISMA-compliant electrical systematic research was performed in the databases including PubMed, Web of Science and Scopus for English publication from their inceptions until January 2024. Data for collection will include primary CAD animal models and any cardiomyocyte cell line with primary hypoxia model. The article title, authors, type of models, signaling pathways and biological changes (proliferation, migration, lumen formation and apoptosis) will be recorded.

**Data availability statement:** All relevant data are within the paper and its Supporting Information files.

**Funding:** This article was funded by Yunnan Fundamental Research Projects, China, Nos. 202201AT070277 and 202102AA310003, High-level Scientific and Technological Talents Program of Yunnan Province, China, No.202305AD160059. Health Science and Technology Project of Kunming, China, No. 2022-SW-011. The funders had supported the decision to publish. The funders had no role in study design, data collection and analysis, or reparation of the manuscript.

**Competing interests:** The authors have declared that no competing interests exist.

## Conclusion

This will provide a new approach in understanding molecular interactions on CAD for new perspective and target treatment for CAD patients in future, especially that intolerance of invasive coronary therapy.

## Registration

Registered in PROSPERO on 10 April, 2024. (CRD42024504245) (https://www.crd.york.ac.uk/prospero/display_record.php?ID=CRD42024504245).

## Introduction

Cardiovascular diseases (CVDs) remain a main cause of death worldwide, posing a serious threat to individual health and burdening healthcare systems [1]. CVDs content myocardial structural changes, coronary vascular lesions, and microcirculatory disturbances, with ischemic heart disease (IHD) being the primary cause of death [1]. As complications caused by CAD progress, distinguishing disease heterogeneity becomes increasingly challenging, highlighting the need for accurate diagnosis and stage evaluation. Identifying molecular mechanisms and regulatory pathways is crucial for early diagnosis and monitoring disease progression in at-risk individuals.

Long non-coding RNA (lncRNA), serving as effective information continuum, play a regulatory role in protein-coding genes with low potential of protein-coding functions [2]. LncRNA, defined as non-translated transcripts exceeding 200 nucleotides [3], typically range in size from approximately 1 kb to over 100 kb [4]. They possess modular structures and function as a class of functional RNA sequences with essential roles in chromosome modification, transcription regulation, protein translation and localization, as well as crucial functions in nuclear transport [2]. Moreover, lncRNA plays important role in cellular processes, development and progression of CVDs, such as cell development, cell adhesion, cell cycle, cell signal transduction, and molecular phenotype modulation [5,6]. Research on lncRNA has become a frontier in CVDs screening and diagnosis as it has shown to participate in the development and progression of CVDs [7,8]. One of the highly conserved lncRNA, MALAT1 plays a regulatory role in early stages of the atherosclerotic development [9]. However, more research indicates that, MALAT1 involve in various pathological and physiological processes by regulating angiogenesis, cell apoptosis, proliferation and autophagy, that plays a role in the formation, occurrence, and progression of CVDs [10–15]. Previous study has shown that MALAT1 upregulates NLCR5 gene overexpression that could block the effect of miR-125b-5p overexpression and significantly reduce hypoxia/reperfusion-induced apoptosis in HL-1 cardiomyocyte cell line [16]. This provides strong evidence for guiding future genetic therapeutic strategies for CAD. Additionally, MALAT1 siRNA can reverse ischemia/reperfusion (I/R) induced upregulation of β-catenin expression, inhibiting myocardial infarction (MI) caused by I/R and improving heart function in rats [17]. Research also suggests that MALAT1 sponges miR-25-3p to upregulate CDC42, participating in the activation of

the MEK/ERK pathway and inhibiting angiogenesis and cardiomyocytes (CMs) regeneration after MI [11]. Study indicates that downregulation of lncMALAT1 can inhibit the ERK/MAPK pathway, thereby significantly improving cardiac function in Sprague-Dawley rats after MI [18].

CMs hypoxia plays a significantly pathological basic role for CAD, while angiogenesis is crucial for microvascular formation in CAD. Various of vascular changes occurs during the process of angiogenesis, which are regulated by several target genes and signaling pathways. The Klotho/FGF23 axis involved in the development of molecular interaction network, while ERK/MAPK and Wnt/β-catenin are essential components of the signaling pathway network structure for cell proliferation, angiogenesis, viability, apoptosis, autophagy, differentiation and migration.

This systematic review aims to evaluate the regulatory role of MALAT1 in CAD through molecular targets and signaling pathways via in vitro and in vivo research models. To our knowledge, the detailed assessment of MALAT1's role and signaling pathways in CAD progression via in vitro and in vivo research models are still limited. Thus, we propose a systematic review protocol following the Preferred Reporting Items for Systematic Reviews and Meta-Analyses Protocols (PRISMA-P) guidelines and the Population, Intervention, Comparison, Outcome, and Study (PICOS) framework, outlining a reproducible strategy for evaluating the effectiveness of literatures. This systematic review protocol outlines the steps to be taken during the review and adheres transparently to the PRISMA guidelines. Furthermore, the results of the systematic review developed based on this protocol can guide future research on the association between MALAT1 and CAD.

## Methods

The development of the systematic review protocol is in accordance with the PRISMA-P [19]. This protocol will help guide the reviewers to answer the following research questions: (1) do lncMALAT1 has roles in CAD? (2) do lncMALAT1 activating upstream signalling Klotho/FGF23 and regulate the downstream Wnt/β-catenin or ERK/MAPK pathway or any other signaling pathways? and (3) what are the potential molecular changes, such as migration, proliferation, lumen formation and apoptosis associated with lncMALAT1? The protocol for this review is registered on the international prospective register of systematic reviews (PROSPERO ID: CRD42024504245). Since this systematic review will extract scientific literature from public databases, ethical approval is not required.

### Inclusion and exclusion criteria

The inclusion and exclusion criteria are divided according to animal and in vitro studies. For animal study, primary CAD animal model established with coronary artery in acute and chronic operation (not limited to coronary artery ligation, ischemia/reperfusion, cryo-/electrical injury, micro-embolism, pharmacological induction, genetic models) of any type, given at any time and diet condition, all sex, all age, and all species will be included in this protocol. CAD model that involved aorta stenosis, heart failure and cardiomyopathy will be excluded.

For in vitro study, any cardiomyocyte cell line (all human and animal) primary hypoxia model (established with low oxygen concentration or cobalt chloride agent) in any level, given at any time, concentration, frequency, and duration. Cardiomyocytes derived from the induced stem cells, and genetically modified cardiomyocytes will be excluded.

### Study intervention and comparators

**Animal study.** The intervention that will be reviewed including CAD animal models exposed to (i) any viral vector for gene therapy (e.g., lentivirus, recombinant adenovirus) in any dosing, given at any time and frequency of dosing; (ii) any types of cells (e.g., cardiac cells, stem cell including bone marrow mesenchymal, adipose mesenchymal, umbilical cord mesenchymal) in any dosing, given at any time and frequency of dosing; (iii) CAD drugs in any dosing, given at any time and frequency of dosing and (iv) device: invasive (coronary stent or scaffolds, coronary balloon catheter) in any material, non-invasive (hyperbaric oxygen) in any oxygen concentration, will be given at any time, duration, and frequency. Further, at least any two of these three groups will be included: (i) Negative control group (placebo, vehicle vector); (ii) Positive

control group (drugs/agents, cells, device) and (iii) Healthy group with or without (iv) Sham group. Any animal model that did not report signalling pathways, vascular changes or without comparator/ control groups necessary for meaningful interpretation of experimental outcomes related to lnc-MALAT1 in CAD will be excluded.

**In vitro study.** The intervention that will be reviewed including any cardiomyocyte cell lines exposed to (i) any viral vector for gene therapy (e.g., lentivirus, recombinant adenovirus) in any concentration, given at any time and frequency of concentration; (ii) cell-derived extracellular vesicles (e.g., exosomes, micro-vesicles or both) in any concentration, given at any time and frequency of concentration with no restriction on method of isolation and purification. (iii) Drugs/agents in any concentration, given at any time and frequency of concentration. Moreover, at least any two of these three groups will be included: (i) Negative control group (placebo, viral vector); (ii) Positive control group (drugs/agents) and (iii) Healthy group. Any in vitro studies utilize cell culture model that did not report signalling pathways, biological changes, without control conditions (normal oxygen concentration of standard cell culturing protocol (37°C, $CO_2$ 5%) or those failing to implement controls essential for accurate assessment of the role and signaling pathway of lnc-MALAT1 in hypoxia progression will be excluded.

## Outcome measures

The primary outcome will be the identification and characterization of Klotho/FGF23 and Wnt/β-catenin and ERK/MAPK pathways for both animal and in vitro studies. These pathways have been chosen as primary outcomes due to play roles in cardiovascular pathophysiology and potential interaction with MALAT1. The secondary outcome will be the identification of other signalling pathways and vascular/biological changes (proliferation, migration, lumen formation, apoptosis), as these processes are crucial in the development and progression of CAD. The efficacy outcome measures will be excluded in both studies.

## Study design

Controlled experimental studies utilizing CAD animal model or in vitro hypoxia condition will be included. Original research articles in English language will be considered. Studies with non-available full text, reviews, systematic reviews, meta-analyses, conference abstracts and non-English language, letter to editors, book, editorial comments, abstract, conference proceedings will be excluded. The eligibility criteria based on the PICOS framework are summarised in Supporting information S1 Table.

## Search strategy and sources

Studies will be identified by searching electronic databases, including Pubmed, Web of Science and Scopus database, from inception until the 01 February 2024. A search strategy incorporating a blend of Medical Subject Headings and keywords together with Boolean operators will be formulated. The planned search strategy for one of the databases (Scopus) is presented in Table 1. Search string: ((MALAT1) OR (lncMALAT1)) AND ((((((("coronary artery disease") OR ("atherosclerosis")) OR ("myocardial infarction")) OR ("unstable angina")) OR ("myocardial bridge")) OR ("myocardial ischemia")). Full search strategies and information for all databases are available in Supporting information S2 Table. The references of eligible articles will also be screened for relevant studies. Gray literature or evidence not published in commercial publications will not be included in this review. Full information of the checklist is available in Supporting information S3 Checklist.

## Study selection

After conducting the search across all specified databases, one researcher (JZ) will import the retrieved studies into End-Note™ 20. Any duplicates will be removed by the duplicated identification feature of EndNote™ 20 before proceeding with the screening of potential studies. The citations gathered through three electronic databases search will be organized in a web-based application Rayyan [20], an AI-powered tool that aids in the identification of studies during a systematic review.

**Table 1. Search strategy in Pubmed, Web of Science and Scopus databases.**

| Search# | Search items |
| --- | --- |
| #1 | MALAT1 (All Fields) |
| #2 | lncMALAT1 (All Fields) |
| #3 | #1 or #2 |
| #4 | "coronary artery disease" |
| #5 | "atherosclerosis" |
| #6 | "myocardial infarction" |
| #7 | "unstable angina" |
| #8 | "myocardial bridge" |
| #9 | "myocardial ischemia" |
| #10 | #4 or #5 or #6 or #7 or #8 or #9 |
| #11 | #3 AND #10 |

To ensure the transparency of the screening process, the justification for exclusion (wrong study design, wrong population, wrong language, wrong comparator, wrong outcome, etc.) will be marked in Rayyan. Screening and selection of articles will be carried out with a step below. In the first step, two researchers (JZ and NFJ) will screen through all titles and abstract from the search results in databases. Researchers will meet up to compare the results and to resolve discrepancies. Next, similarly, JZ and NFJ will be working independently to screen through the full article obtained from the second step. Efforts will be made to obtain the full papers through the university library if full papers are not available online. The full articles will be studied through to ensure that they meet the objectives. Researchers will meet upon completion to compare results. Any discrepancies will be resolved with the other researchers (AFML and NFR) through discussion or voting, which contribute to improve transparency in this process. To summarize the study selection process, including the number of citations retrieved using our search terms, the final count of included studies, and the reasons for excluding studies, a PRISMA flow diagram (Fig 1) [21] will be employed.

## Data management and extraction

A comprehensive, standardized data extraction form will be developed using Microsoft Excel and piloted for variable extraction by two independent reviewers (JZ and NFJ). The identified citations and extracted data will be regularly backed up to prevent data loss. Demographic data will include the first author's name, year of publication, country, study design, and experimental study used to form an initial development. Two reviewers (JZ and NFJ) will pilot the form on a random sample of 20% studies, which representing diverse study designs, that meet the inclusion criteria. The reviewers will refine the form accordingly while identified by any difficulty after discussion. The revised form will be tested on an additional 10% studies to address all concerns. The third independent reviewer (AFML) will provide any additional revision after reviewing the piloted form before finalization. Data from the animal model will be extracted including number of CAD and control groups, and animal model with (i) animal species, sex, age, weight; (ii) CAD model induction method and anesthesia method. For in vitro study, the data that will be included including number of hypoxia and control groups, types of cardiomyocytes, passage number; hypoxia model induction, duration, concentration and frequency. The primary outcomes will be extracted include the fold change of up and down regulation Klotho/FGF23 and Wnt/β-catenin and ERK/MAPK expressions in both studies. The data will be quantitative. Secondary outcome will be extracted include other signalling pathways - up and down regulation (fold change) and vascular or biological changes (proliferation - cell percentage, migration - cell distance in micrometer, lumen formation - number of branches, apoptosis - cell death of percentage) in both studies respectively. Additionally, details on the bibliographical, such as author, year and language in both animal and in vitro studies will also be extracted. A standardised data extraction will be developed and piloted for variable extraction by two

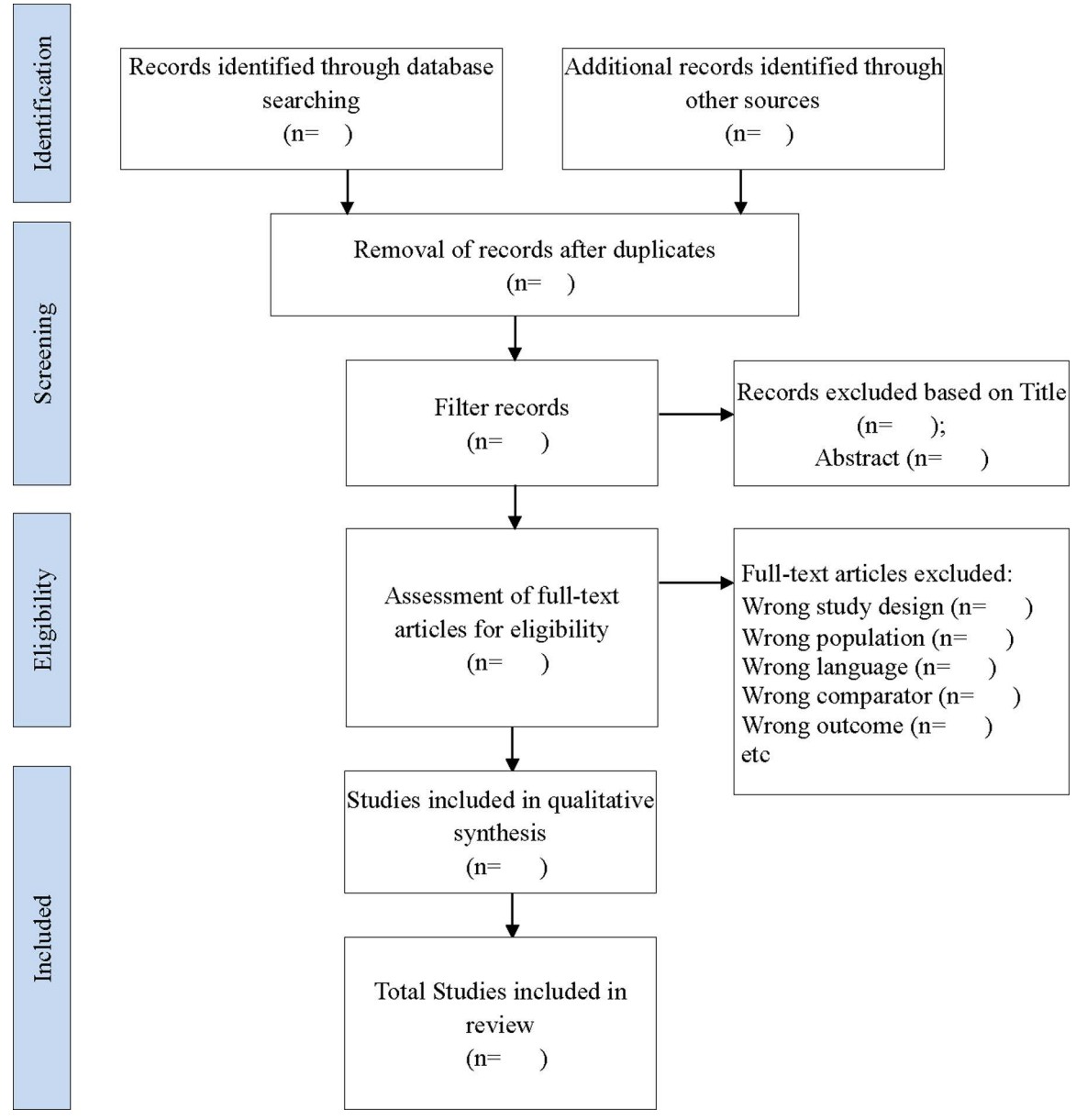

**Fig 1. PRISMA flow diagram for systematic reviews.**

independent reviewers (JZ and NFJ) for table tabulation. Duplicated studies will be excluded. Any significant differences in data extraction will be resolved through discussion between the reviewers. If necessary, a third reviewer (AFML) will be consulted. To ensure data quality and consistency, a random sample of 30% of the included studies will be cross-checked by a third reviewer (AFML). If significant differences are found, a full review of all extracted data will be conducted.

## Quality assessment and bias assessment

The quality assessment on the included publications was carried out using the SYRCLE's risk of bias tool [22]. The criteria including: i) were inclusion/ exclusion criteria reported; ii) was the search adequate; iii) were the included studies

synthesized; iv) was the quality of the included studies assessed; v) are sufficient details about the individual included studies presented. Randomization and heterogeneity in animal and in vitro studies provide a more accurate reflection of reality. In addition, the proposed random effects model present a high level of evidence [23]. For each domain, the reviewers will make a judgment to the risk of bias as 'low', 'high', or 'unclear'. If a study to be included with a 'low' risk of bias judgment, at least four criteria must be meet. The process of the quality assessment on included studies will be evaluated by independent reviewers (JZ, NFJ and AFML). To minimize selection bias, we will employ independent screening of titles, abstracts, and full texts. Any disagreements will be resolved through discussion or consultation by a fourth reviewer (NFR). Considering study limitations, incomplete outcome, imprecision, and publication bias, any discrepancies or disagreements, a fourth reviewer (NFR) will be consulted if possible.

### Data synthesis

For data synthesis, we will employ a narrative synthesis approach. All extracted data from eligible studies will be exactly recorded into Microsoft Excel with sufficient information by dividing into animal model and in vitro model two parts. The synthesized data will be presented in tabular or graphical form.

### Planned statistical methods

Data synthesis will employ a narrative approach to summarize the findings. Quantitative data (e.g., animal age, CAD duration, infarct size, apoptosis rates) will be categorized by species and gene expression (e.g., upregulation and downregulation). For studies reporting continuous data (e.g., migration distance), we will calculate mean ± standard deviation with 95% confidence intervals (CIs) where comparable metrics are available. Heterogeneity in experimental designs will be addressed through subgroup analysis (e.g., animal vs. in vitro models, gene expression). Trends will be summarized using descriptive statistics (e.g., frequency of pathway activation). P-value < 0.05 was considered significant. All analyses will be performed in Excel and GraphPad Prism, with results presented in both tabular and graphical form. Graphics will be visualized through Adobe Illustrator.

### Expected results

This systematic review aims to provide strong evidence supproting the hypothesis that MALAT1 is involved in the pathophysiological processes of CAD in animal and in vitro studies. For animal studies, it will gather molecular, cellular, or physiological evidence on how MALAT1 potentially involved in progression of CAD. Additionally, it will complement existing evidence based on the Klotho/FGF23 axis and Wnt/β-catenin and ERK/MAPK as well as other signaling pathways, as potential molecular targets and vascular/biological changes, such as endothelial cell dysfunction, inflammatory cell activation, and etc. By synthesizing evidence of these studies, this protocol will not only strengthen the hypothesis of lncRNA involved in the pathophysiological of CAD, but also provide a method to overview the literature in the field of CAD and lncRNA biology.

### Discussion

MALAT1, as a tumor-associated lncRNA, primarily participates in gene expression splicing and epigenetic regulation [24]. A latest report suggests that MALAT1 acts as a regulatory factor in CVDs [8] and elevated levels of MALAT1 expression indicate poor prognosis in patients with acute coronary syndrome (ACS) [25]. Meanwhile, in vitro studies showed that the expression of MALAT1 significantly increased in H9C2 cells after hypoxia treatment while MALAT1 knockdown aggravated hypoxia-induced damage in H9C2 cells by enhancing cell viability, migration, invasion inhibition, and promoting cell apoptosis [26]. From these evidence, the regulation of MALAT1 intracellular signaling pathways is diverse and highly complex as the same molecule may have different regulatory effects in different models. The differential effects of MALAT1 on CMs

proliferation and apoptosis may be related to different signaling pathways involved. As a protocol for systematic review, it will summarize the literatures on the role and signaling pathways of lncMALAT1 in CAD by animal and in vitro studies to provide evidences of target therapeutic in future.

More pathways can also be involved in the biological changes of myocardial microcirculation with CAD. MALAT1 exhibited a significant association with the development of CAD and decreased expression in atherosclerotic plaques [27]. Moreover, MALAT1 involved in the regulatory of upstream and downstream signalling. In the upstream pathway, MALAT1 epigenetically inhibits the expression of Klotho, promotes cell apoptosis, reduces cell proliferation, and then induces glomerular endothelial cell damage [28]. In the downstream pathway, MALAT1 knockdown inhibits the Wnt/β-catenin pathway based on miRNA-based microarray analysis and bioinformatics prediction [29], which leads to decrease cardiomyocyte apoptosis, reduces the level of pro-inflammatory factors, and attenuates myocardial tissue injury [17]. Moreover, MALAT1 activates the MEK/ERK pathway and inhibits angiogenesis and myocardial regeneration after myocardial infarction [11]. However, down-regulation of MALAT1 to improve cardiac function after myocardial infarction may also be achieved by inhibiting ERK/MAPK [18].

Typically, in vitro studies focus on specific cellular mechanisms and present particular limitations. Animals share many physiological and genetic similarities with humans, and animal studies can be more precisely validated by in vivo disease models. Physiological mechanisms are influenced by several factors depending on the integral functional properties, although it is contributed to achieve more precisely how substances or interventions affect the process of disease development, such as multiple genetic material, different pathways and uncontrolled experimental conditions. In summary, investigating animal and in vitro studies together worth to bridge the gap between both, provide a more comprehensive understanding of biological processes, contribute to the assessment of the relevance of findings to clinical disease, and assist in the development of effective interventions and therapeutic approaches. Moreover, one of the strengths of this study is the adoption of an open and reproducible method for systematically reviewing the literature. The results revealed from this review will contribute positively to clarifying the relationship between lncRNA and CAD, understanding the target moleculars and mechanisms of action, corresponding changes in myocardial microcirculatory structure, and guiding molecular targeted therapy for CAD in clinical practice in future.

## Supporting information

**S1 Table. Population, Intervention, Comparison, Outcome and Study framework (PICOS).**
(DOCX)

**S2 Table. Search strategies and results.**
(DOCX)

**S3 Checklist. PRISMA P checklist.**
(DOCX)

## Author contributions

**Conceptualization:** Jia Zheng, Arimi Fitri Mat Ludin, Nor Fadilah Rajab, Nurul Farhana Jufri.

**Funding acquisition:** Jia Zheng.

**Investigation:** Jia Zheng, Arimi Fitri Mat Ludin, Nor Fadilah Rajab, Li Shaolong, Nurul Farhana Jufri.

**Methodology:** Jia Zheng, Arimi Fitri Mat Ludin, Nor Fadilah Rajab, Nurul Farhana Jufri.

**Software:** Jia Zheng.

**Supervision:** Arimi Fitri Mat Ludin, Nor Fadilah Rajab, Li Shaolong, Nurul Farhana Jufri.

**Writing – original draft:** Jia Zheng.

**Writing – review & editing:** Arimi Fitri Mat Ludin, Nor Fadilah Rajab, Nurul Farhana Jufri.

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
