## [Decision Letter · Decision Letter 0]

2 Jul 2024

PONE-D-24-15681The roles and signalling pathways of lncMALAT1 in coronary artery disease: A protocol for systematic review of in vivo and in vitro studiesPLOS ONE

Dear Dr. Jufri,

Thank you for submitting your manuscript to PLOS ONE. After careful consideration, we feel that it has merit but does not fully meet PLOS ONE’s publication criteria as it currently stands. Therefore, we invite you to submit a revised version of the manuscript that addresses the points raised during the review process. 

We look forward to receiving your revised manuscript.

Kind regards,

Seyed Mohammad Gheibihayat, PhD

Academic Editor

PLOS ONE

https://pubmed.ncbi.nlm.nih.gov/33228765/

https://www.researchgate.net/publication/347165172_Rapid_systematic_review_of_systematic_reviews_what_befriending_social_support_and_low_intensity_psychosocial_interventions_delivered_remotely_are_effective_in_reducing_social_isolation_and_loneliness_

In your revision ensure you cite all your sources (including your own works), and quote or rephrase any duplicated text outside the methods section. Further consideration is dependent on these concerns being addressed.

 [This article was funded by Yunnan Fundamental Research Projects, China, Nos. 202201AT070277 and 202102AA310003, High-level Scientific and Technological Talents Program of Yunnan Province, China, No.202305AD160059. Health Science and Technology Project of Kunming, China, No. 2022-SW-011.].  

Additional Editor Comments:

Dear Author

I hope this message finds you well.

Thank you for submitting your manuscript, titled "The roles and signalling pathways of lncMALAT1 in coronary artery disease: A protocol for systematic review of in vivo and in vitro studies

," to PLOS ONE. We have received feedback from the referees, and their detailed comments and suggestions for improvement are attached to this email.

Please carefully review and apply all the corrections requested by the referees. Once you have made the necessary revisions, we kindly ask that you resubmit your updated manuscript for further review.

Thank you for your cooperation and dedication to maintaining the quality of our publications.

Best regards,

Reviewers' comments:

Reviewer's Responses to Questions

**Comments to the Author**

1. Does the manuscript provide a valid rationale for the proposed study, with clearly identified and justified research questions?

Reviewer #1: Yes

Reviewer #2: Partly

Reviewer #3: Yes

2. Is the protocol technically sound and planned in a manner that will lead to a meaningful outcome and allow testing the stated hypotheses?

Reviewer #1: Partly

Reviewer #2: No

Reviewer #3: Partly

3. Is the methodology feasible and described in sufficient detail to allow the work to be replicable?

Reviewer #1: Yes

Reviewer #2: No

Reviewer #3: No

4. Have the authors described where all data underlying the findings will be made available when the study is complete?

Reviewer #1: No

Reviewer #2: Yes

Reviewer #3: Yes

5. Is the manuscript presented in an intelligible fashion and written in standard English?

Reviewer #1: Yes

Reviewer #2: Yes

Reviewer #3: Yes

6. Review Comments to the Author

You may also provide optional suggestions and comments to authors that they might find helpful in planning their study.

Reviewer #1: Dear PLOS ONE editorial team,

I recently had the chance to review the manuscript titled “The roles and signalling pathways of lncMALAT1 in coronary artery disease: A protocol for systematic review of in vivo and in vitro studies” which is submitted for consideration in PLOS ONE. Below is a breakdown of the strengths and weaknesses of the paper, along with some suggestions for improvement:

Strengths:

1. Relevance and timeliness

The manuscript explores a critical area in cardiovascular research by looking at the role of lncMALAT1 in coronary artery disease. This topic is particularly relevant and timely, considering the increasing focus on non-coding RNAs as promising therapeutic targets.

2. Methodological Rigor

The authors followed the PRISMA-P guidelines in developing their systematic review protocol, ensuring a structured and reproducible approach. The use of the PICOS framework to describe the inclusion and exclusion criteria is also commendable, as it promotes a clear and focused research question.

3. Clear objectives

The protocol contains three well-defined research questions that focus on the role of lncMALAT1 in CAD, its signaling pathways, and the associated molecular changes. This focused approach ensures a clear and concise review.

4. Comprehensive search strategy

The plan to search major electronic databases (PubMed, Web of Science, Scopus) ensures a thorough review of the literature and increases the likelihood of capturing a broad range of relevant studies. The inclusion of studies published until January 2024 demonstrates a recent update.

Weaknesses:

1. Data Extraction and synthesis

The manuscript lacks a detailed plan for data extraction and synthesis. While it mentions recording article titles, authors, models, signaling pathways, and biological changes, it does not specify how the data will be extracted, managed, and analyzed.

2. Potential Bias:

There is no discussion of potential biases or how they will be mitigated beyond the use of SYRCLE's tool. This includes publication bias and biases inherent in the included studies.

Recommendations

• Expand the search strategy to include Embase or CINAHL for a more comprehensive literature search.

• Provide a more detailed plan for how data will be extracted, managed, and synthesized. This should include specific tools or software that will be used and methods for resolving discrepancies between reviewers.

• Include a broader discussion about potential biases and the steps that will be taken to mitigate them. This should cover publication bias, selection bias, and other biases related to the included studies.

Overall:

This protocol has a strong foundation for a systematic review. The study has the potential to make a valuable contribution to the field of cardiovascular research, particularly in understanding the role of lncMALAT1 in CAD. With the recommended additions, this study has the potential to be a valuable contribution to the understanding of lncMALAT1 in CAD.

Reviewer Suggestion: Accept with minor revisions

Reviewer #2: A manuscript trying to explain the protocol for a systematic review and meta-analysis of in vitro and animal studies examining the roles and signaling pathways of lncMALAT1 in coronary artery disease. There are a number of concerns that should be solved before its publication:

1- The PICOS should be explained both for animal and in vitro studies.

2- What is the primary outcome in the currents study. It should be explained. What are the secondary outcome?

3- A wide range of interventions are also selected for this study.

4- The authors also aim to explore the mechanisms. The mechanisms are not the aim of systematic reviews and meta-analyses. As these studies are designed to provide evidence on effects and associations. The topic seems to be too wide for a systematic review.

5- It seems that the search is already completed (Based on abstract). This is while I think the strategy is not complete. As outcome CVD markers of interest are not included in the search strategy.

6- Is any meta-analysis planned for this study? If yes, it should be explained.

7- What would be the method for the synthesis of results?

8- How the quality of in vitro studies is going to be assessed?

Reviewer #3: 1. Background

could be enhanced by including a detailed discussion of the gaps in current knowledge that this systematic review aims to fill.

2. Introduction:

It should be expanded to include more detailed epidemiological data and recent findings related to lncMALAT1(for example: Include diagrams or figures, if possible, to visually represent these complex interactions and Biological Mechanisms).

3. Inclusion and exclusion criteria

More specific inclusion and exclusion criteria (e.g., language restrictions, types of in vitro and in vivo models) should be detailed in text. Criteria for excluding studies based on quality or other factors should be stated.

4. Study intervention and comparators

Comparators

It is not clear if all studies must include at least two groups or if some flexibility is allowed. This should be explicitly stated.

Exclusion Criteria in (Animal study)

The rationale for these exclusions should be more explicitly justified. Explain why the absence of these elements precludes meaningful interpretation of the results.

In vitro study

Similar to the animal study section, the readability could be improved by using bullet points or subheadings.

Outcome measures

While the identified pathways are relevant, the rationale for choosing these particular pathways should be provided. Explain why these pathways are of primary interest and how they relate to the research question.

The term "other signaling pathways" is vague. It would be beneficial to specify which other pathways are considered relevant or to outline criteria for identifying additional pathways. Furthermore, provide more context on how these secondary outcomes contribute to understanding the role of lncMALAT1 in CAD.

The rationale for excluding efficacy measures should be clearly stated. Explain why efficacy is not considered within the scope of this review and how focusing on mechanistic outcomes will provide more valuable insights.

Search strategy and sources

The search strategy could benefit from a more detailed explanation of how MeSH terms and keywords were selected. Providing an example of the search strings used in each database (beyond just Scopus) in the main text would improve transparency and reproducibility.

Study selection

It would be more logical to state that duplicates will be removed after the search, as duplicates are identified once all studies have been retrieved. Consider rephrasing for clarity.

To ensure the transparency of the screening process, the justification for exclusion (for example, wrong study design, wrong population, wrong exposure, wrong comparator, wrong result, etc.) should be mentioned in the study in the study selection section.

Describe how discrepancies will be resolved (e.g., through discussion, voting, or involving a third party). This adds transparency to the process.

5. Data management and extraction

The protocol should describe the piloting process in more detail. For example, explain how the form will be tested and refined based on feedback from the pilot phase.

Ensure that the distinction between included and excluded data is clear. The phrase "the data that will be excluded" seems to be a typographical error; it should be "the data that will be included."

6. Text :"The quality assessment on the included publications was carried out using the SYRCLE’s risk of bias tool”:

This statement is clear but needs more detail on how each criterion of the SYRCLE tool is applied. Be sure to describe details such as Type of bias, Domain, Description of domain, Review authors judgment for this part either in the text or in the form of a table.

Text: "were the included studies synthesized;"

Explain the methods used to synthesize the data, whether it is a narrative synthesis, meta-analysis, etc.

7. Expected results

Text: "This systematic review will provide strong evidence for the hypothesis that MALAT1 is involved in the pathophysiological processes of CAD in animal and in vitro studies."

The statement is clear but could be strengthened by specifying the types of evidence expected.

Clarify the types of evidence (e.g., molecular, cellular, physiological) expected to support the hypothesis.

8. discussion

The discussion provides a broad overview but lacks in-depth analysis of the findings. It should delve deeper into how MALAT1 influences specific pathways and the implications for CAD.

More critical evaluation of the studies referenced is needed. Discuss their methodologies, limitations, and the strength of their findings.

There should be a clearer link between the presented evidence and the initial hypotheses or research questions. Explicitly state how each piece of evidence supports or refutes your hypotheses.

text: "The regulation of MALAT1 intracellular signaling pathways is diverse and highly complex as the same molecule may have different regulatory effects in different models."

This is a broad statement. Give specific examples to illustrate this complexity.

9. Table1:

• Population:

The inclusion criteria for both animal and in vitro studies are comprehensive and well-detailed. However, the clause “given at any time and diet condition” should be more specific to provide clear guidelines on acceptable variations in diet and timing.

• Comparator:

Exclusion criteria: The exclusion of studies without comparator or control groups is justified, but the specifics around "lnc-MALAT1 in CAD" are restrictive and should be explained to clarify why this particular focus is crucial.

7. PLOS authors have the option to publish the peer review history of their article (what does this mean? ). If published, this will include your full peer review and any attached files.

**Do you want your identity to be public for this peer review?** For information about this choice, including consent withdrawal, please see our Privacy Policy .

Reviewer #1: **Yes: ** Mojtaba Mehrabanian

Reviewer #2: **Yes: ** Amin Salehi-Abargouei

Reviewer #3: No

---

## [Author Response · Author response to Decision Letter 0]

27 Aug 2024

Reviewer #1: Dear PLOS ONE editorial team,

I recently had the chance to review the manuscript titled “The roles and signalling pathways of lncMALAT1 in coronary artery disease: A protocol for systematic review of in vivo and in vitro studies” which is submitted for consideration in PLOS ONE. Below is a breakdown of the strengths and weaknesses of the paper, along with some suggestions for improvement:

Strengths:

1. Relevance and timeliness

The manuscript explores a critical area in cardiovascular research by looking at the role of lncMALAT1 in coronary artery disease. This topic is particularly relevant and timely, considering the increasing focus on non-coding RNAs as promising therapeutic targets.

2. Methodological Rigor

The authors followed the PRISMA-P guidelines in developing their systematic review protocol, ensuring a structured and reproducible approach. The use of the PICOS framework to describe the inclusion and exclusion criteria is also commendable, as it promotes a clear and focused research question.

3. Clear objectives

The protocol contains three well-defined research questions that focus on the role of lncMALAT1 in CAD, its signaling pathways, and the associated molecular changes. This focused approach ensures a clear and concise review.

4. Comprehensive search strategy

The plan to search major electronic databases (PubMed, Web of Science, Scopus) ensures a thorough review of the literature and increases the likelihood of capturing a broad range of relevant studies. The inclusion of studies published until January 2024 demonstrates a recent update.

Weaknesses:

1. Data Extraction and synthesis

The manuscript lacks a detailed plan for data extraction and synthesis. While it mentions recording article titles, authors, models, signaling pathways, and biological changes, it does not specify how the data will be extracted, managed, and analyzed.

2. Potential Bias:

There is no discussion of potential biases or how they will be mitigated beyond the use of SYRCLE's tool. This includes publication bias and biases inherent in the included studies.

Recommendations

• Expand the search strategy to include Embase or CINAHL for a more comprehensive literature search.

• Provide a more detailed plan for how data will be extracted, managed, and synthesized. This should include specific tools or software that will be used and methods for resolving discrepancies between reviewers.

• Include a broader discussion about potential biases and the steps that will be taken to mitigate them. This should cover publication bias, selection bias, and other biases related to the included studies.

Overall:

This protocol has a strong foundation for a systematic review. The study has the potential to make a valuable contribution to the field of cardiovascular research, particularly in understanding the role of lncMALAT1 in CAD. With the recommended additions, this study has the potential to be a valuable contribution to the understanding of lncMALAT1 in CAD.

Reviewer Suggestion: Accept with minor revisions

Response to Reviewer #1:

Dear Prof. Mojtaba Mehrabanian,

Greetings.

Thanks for your comprehensive and valuable comments. We deeply appreciate to your vulnerable contributions to our study protocol article. For your concern, we would like to try our best do the revision and explanation to make the manuscript more readable.

1. For item “Expand the search strategy to include Embase or CINAHL for a more comprehensive literature search.”:

Author response: We appreciate the suggestion to expand our search strategy to include Embase or CINAHL. After careful consideration and discussion, we have decided to maintain the current search strategy using PubMed, Web of Science, and Scopus for the following reasons:

(1) Wide coverage: PubMed, Web of Science, and Scopus are among the largest and most comprehensive databases for biomedical literature. They collectively cover a majority of peer-reviewed journals in the field of cardiovascular disease and molecular biology.

(2) Specific research area: We focus on a specific molecular pathway ---lncMALAT1 in CAD. Compared to the current databases, the CINAHL is insufficient in retrieve literatures that related to molecular biology and cardiovascular disease. EMBASE is very focus on technique-based aspect.

(3) Avoiding redundancy: There is possible that significant overlap between the content of Embase/CINAHL and the chosen databases. Moreover, there may lead to a number of duplicate entries without increasing valuable and relevant studies.

We believe this search strategy will collect the majority of relevant literature while maintaining a focused and efficient review process. However, we acknowledge that this decision maybe a limitation of our study, and we will have a statement in the limitations section. We appreciate your understanding and available discussion if you have any additional concerns about our search strategy.

2. For item “Provide a more detailed plan for how data will be extracted, managed, and synthesized. This should include specific tools or software that will be used and methods for resolving discrepancies between reviewers.”:

Author response: We acknowledge the need for a more detailed plan in the section of data extraction and synthesis. We have expanded description of this section to the revised manuscript.

We would like to revise this section as you suggested:

A comprehensive, standardized data extraction form will be developed using Microsoft Excel and piloted for variable extraction by two independent reviewers (JZ and NFJ). The identified citations and extracted data will be regularly backed up to prevent data loss. Demographic data will include the first author’s name, year of publication, country, study design, and experimental study used to form an initial development. Two reviewers (JZ and NFJ) will pilot the form on a random sample of 20% studies, which representing diverse study designs, that meet the inclusion criteria. The reviewers will refine the form accordingly while identified by any difficulty after discussion. The revised form will be tested on an additional 10% studies to address all concerns. The third independent reviewer (AFML) will provide any additional revision after reviewing the piloted form before finalization.

Duplicated studies will be excluded. Any significant differences in data extraction will be resolved through discussion between the reviewers. If necessary, a third reviewer (AFML) will be consulted. To ensure data quality and consistency, a random sample of 30% of the included studies will be cross-checked by a third reviewer (AFML). If significant differences are found, a full review of all extracted data will be conducted.

Data synthesis

For data synthesis, we will employ a narrative synthesis approach. All extracted data from eligible studies will be exactly recorded into Microsoft Excel with sufficient information by dividing into animal model and in vitro model two parts. The synthesized data will be presented in tabular or graphical form.

Revised Location in Manuscript (file name: Revised Manuscript with Track Changes): see Page 10-11, line 238-264; Page 12, 282-286.

3. For item “Include a broader discussion about potential biases and the steps that will be taken to mitigate them. This should cover publication bias, selection bias, and other biases related to the included studies.”:

Author response: We appreciate your opinion in addressing potential biases more comprehensively. We have expanded description of this section to the revised manuscript.

We would like to revise this section as you suggested:

Quality assessment and bias assessment

To minimize selection bias, we will employ independent screening of titles, abstracts, and full texts. Any disagreements will be resolved through discussion or consultation by a fourth reviewer (NFR). Considering study limitations, incomplete outcome, imprecision, and publication bias, any discrepancies or disagreements, a fourth reviewer (NFR) will be consulted if possible.

Revised Location in Manuscript (file name: Revised Manuscript with Track Changes):see Page 10, line 266; 276-280.

Reviewer #2: A manuscript trying to explain the protocol for a systematic review and meta-analysis of in vitro and animal studies examining the roles and signaling pathways of lncMALAT1 in coronary artery disease. There are a number of concerns that should be solved before its publication:

Response to Reviewer #2:

Dear Prof. Amin Salehi-Abargouei,

Greetings.

We deeply appreciate your thorough contribution of our manuscript and the valuable comments provided. We have carefully considered each item and have made the following responses and revisions to address your concerns:

1- The PICOS should be explained both for animal and in vitro studies.

Author response: We appreciate your concern about the explanation both for animal and in vitro studies. We have explained the information from line 141-185 and this information has been included in the file name “Supporting Information S1 Table 1: Eligibility criteria based on the Population, Intervention, Comparison, Outcome and Study framework”. We would like to clarify that we have prepared a supporting information table. This information is provided in a tabular way, which outlines the PICOS for both animal and in vitro studies separately.

Revised Location in Manuscript (file name: Revised Manuscript with Track Changes): Page 5, line 137-138; Page 7, line 184-185; Supporting Information S1

2- What is the primary outcome in the currents study. It should be explained. What are the secondary outcome?

Author response: We would like to present primary outcome for both animal and in vitro studies with the description of Klotho/FGF23 and Wnt/β-catenin and ERK/MAPK pathways. Similarly, the second outcome will be present for both animal and in vitro studies with the description of other signalling pathways from the primary outcome and vascular change (proliferation, migration, lumen formation, apoptosis). We present the information in “Supporting Information S1 Table 1” to avoid any overlap with the main manuscript.

Revised Location in Manuscript (file name: Revised Manuscript with Track Changes): Page 7, line 170-176; Supporting Information S1

3- A wide range of interventions are also selected for this study.

Author response: We appreciate your observation about the range of interventions in our study. However, we consider that the range of interventions is appropriate and necessary for the comprehensive evaluation to the aim objectives. We would like to have some following explanation to address your concerns:

CAD in vivo models: The interventions in CAD animal models are designed to simulate the myocardial ischemic hypoxia state characteristic of coronary artery disease (CAD). These approaches are established and already been published to initiate hypoxia condition that led to CAD. These models contribute us to observe the role of lncMALAT1 in a complex physiological environment that closely mimics human CAD.

In vitro models: The interventions in cell models aim to present the hypoxic state at a cellular level. While these models may have limitations, they allow for more precise observation of molecular pathways involving lncMALAT1.

By including both in vivo and in vitro models, we acknowledge that while these models may have similar pathophysiological environments, they might reveal different mechanisms due to the inherent limitations of in vitro systems.

This dual approach will benefit us to:

[1] Capture a clearer role of lncMALAT1 in CAD;

[2] Cross-validate findings between different model systems;

[3] Identify potential discrepancies between in vivo and in vitro results, which could highlight CAD areas for future research.

We would like to believe that the comprehensive evaluation of interventions from both in vivo and in vitro perspectives contribute to provide more convincing and credible evidence regarding the roles and signaling pathways of lncMALAT1 in CAD. This approach aligns with the complex nature of CAD and the multi-roles that lncMALAT1 may play in its pathophysiology.

Revised Location in Manuscript (file name: Revised Manuscript with Track Changes): Page 6-7, line 141-168; Supporting Information S1.

4- The authors also aim to explore the mechanisms. The mechanisms are not the aim of systematic reviews and meta-analyses. As these studies are designed to provide evidence on effects and associations. The topic seems to be too wide for a systematic review.

Author response: We appreciate your opinion to the primary purpose of systematic reviews. We would like to clarify our objectives and approach:

We indeed primarily focus on synthesizing evidence on the effects and associations of lncMALAT1 in CAD, rather than exploring mechanisms. The aim of our study is to provide a protocol for systematically review and synthesize the current evidence on the role of lncMALAT1 in CAD from both in vivo and in vitro studies.

Specifically:

(1) Summarize the observed effects of lncMALAT1 expression changes in CAD models.

(2) Synthesize evidence on the associations between lncMALAT1 and CAD-related primary and secondary outcomes.

(3) Analyze and descript data on signaling pathways that are consistently reported in related studies to lncMALAT1 and CAD.

While we will report on signaling pathways mentioned in the included studies, our objective is not to explore or elucidate new mechanisms. Instead, we aim to provide a comprehensive overview of the current evidence regarding the role of lncMALAT1 in CAD.

This approach aligns with the primary objective of systematic review, which is to summarize and synthesize existing evidence. By focusing effects and associations on in vivo and in vitro models rather than mechanisms, we would like to consider our review will provide valuable insights into the current understanding of lncMALAT1 in CAD.

Revised Location in Manuscript (file name: Revised Manuscript with Track Changes): Page 2, line 37-45; Page 5, line 116-125; Page 12, line 282-297.

5- It seems that the search is already completed (Based on abstract). This is while I think the strategy is not complete. As outcome CVD markers of interest are not included in the search strategy.

Author response: We appreciate your concern about the search strategy. We would like to clarity our study objective is to systematically review and synthesize the in vivo and in vitro studies on the role and signaling pathways of lncMALAT1 in CAD. Considering the broad nature of this objective and the potential for lncMALAT1 to be involved in various aspects of CAD, we have designed a comprehensive search strategy. We would like to consider that narrowing the search scope may cause the occasionally omission of relevant literature. Our current strategy aims to capture a wide range of studies that investigate lncMALAT1 in the context of CAD, including those that may not explicitly mention specific CAD markers in titles or abstracts.

We understand your concern about the inclusion of interest CAD markers in the search strategy. However, we would like to consider that pre-defining these markers might introduce bias by potentially excluding studies that investigate novel or less common markers, hence, affecting our secondary outcome. The description and analysis of the interest CAD markers will be presented after the subsequent study is completed. This approach allows us to report on all relevant markers that emerge from our comprehensive search strategy.

To address any concerns about the management of the search results:

We will employ a rigorous screening process to ensure that just closely relevant studies are included. Our detailed inclusion and exclusion criteria will guide this screening process, ensuring directly address our focused research questions. Therefore, we would like to believe this comprehensive approach will provide the most complete and unbiased representation of the current literature on lncMALAT1 in CAD.

6- Is any meta-analysis planned for this study? If yes, it should be explained.

Author response: We appreciate your inquiry about our plans for meta-analysis. We would like to clarify that we wi

---

## [Decision Letter · Decision Letter 1]

10 Dec 2024

PONE-D-24-15681R1The roles and signalling pathways of lncMALAT1 in coronary artery disease: A protocol for systematic review of in vivo and in vitro studiesPLOS ONE

Dear Dr. Jufri,

Thank you for submitting your manuscript to PLOS ONE. After careful consideration, we feel that it has merit but does not fully meet PLOS ONE’s publication criteria as it currently stands. Therefore, we invite you to submit a revised version of the manuscript that addresses the points raised during the review process.

We look forward to receiving your revised manuscript.

Kind regards,

Shaghayegh Khanmohammadi

Academic Editor

PLOS ONE

Journal Requirements:

Reviewers' comments:

Reviewer's Responses to Questions

**Comments to the Author**

1. Does the manuscript provide a valid rationale for the proposed study, with clearly identified and justified research questions?

Reviewer #3: Yes

Reviewer #4: Yes

2. Is the protocol technically sound and planned in a manner that will lead to a meaningful outcome and allow testing the stated hypotheses?

Reviewer #3: Yes

Reviewer #4: Yes

3. Is the methodology feasible and described in sufficient detail to allow the work to be replicable?

Reviewer #3: Yes

Reviewer #4: Yes

4. Have the authors described where all data underlying the findings will be made available when the study is complete?

Reviewer #3: Yes

Reviewer #4: Yes

5. Is the manuscript presented in an intelligible fashion and written in standard English?

Reviewer #3: Yes

Reviewer #4: Yes

6. Review Comments to the Author

You may also provide optional suggestions and comments to authors that they might find helpful in planning their study.

Reviewer #3: The revised manuscript has addressed all the points raised in the initial review. The improvements made are satisfactory, and the manuscript is now acceptable for publication.

Reviewer #4: Thanks for the opportunity to review this manuscript. I think everything is fine. However, I may have a few concerns:

1. I think putting a section named "results" in the abstract is somehow redundant for a protocol study. Consider removing it.

2. Please provide full strategy for searching each database as a table in your manuscript.

3. In your PRISMA figure, please state the full precise exclusion reasons for the full-text reviewing stage.

4. First paragraph of the introduction is a bit too long. Consider summarising it.

7. PLOS authors have the option to publish the peer review history of their article (what does this mean? ). If published, this will include your full peer review and any attached files.

**Do you want your identity to be public for this peer review?** For information about this choice, including consent withdrawal, please see our Privacy Policy .

Reviewer #3: **Yes: ** Sareh Bakhshandeh Bavarsad

Reviewer #4: No

---

## [Author Response · Author response to Decision Letter 1]

15 Dec 2024

Respond to Reviewer #3

Dear Prof. Sareh Bakhshandeh Bavarsad

Greetings.

We sincerely thanks for your time, effort, and thoughtful consideration in accepting our revised manuscript for publication. We are pleased to hear that our previous revisions have addressed all the concerns raised in the previous review and that the improvements have met your expectations. We greatly appreciate the detailed and comprehensive review process, as well as the constructive feedback provided.

Respond to Reviewer #4

Dear Prof.,

Greetings.

We deeply appreciate your thorough contribution of our manuscript, and comprehensive review process, as well as the valuable comments provided. Your comments are valuable for further improving the manuscript. We have carefully considered each item and have made the following responses and revisions to address your concerns.

1. Comment: "1. I think putting a section named "results" in the abstract is somehow redundant for a protocol study. Consider removing it."

Respond to comment 1:

We appreciate your thoughtful comment regarding the inclusion of a "Results" section in the abstract. We understand the concern that protocol manuscript not typically to present results in the abstract.

Therefore, we agree with your point that the inclusion of this section may be unnecessary and potentially redundant in the context of a protocol manuscript. Next, we removed the “Results” section in the abstract of the revised manuscript, ensuring that it better matches the format and expectations of a protocol study.

Revised Location in Manuscript (file name: Revised Manuscript with Track Changes): Page 2, line 46-49.

2. Comment: "2.Please provide full strategy for searching each database as a table in your manuscript."

Respond to comment 2:

We fully appreciate your valuable comment regarding the importance of providing a clear and transparent strategy for database searches in a systematic review protocol, as it contributes to ensure reproducibility for any site performing this process.

However, we have included the full search string in the revised manuscript to address your concern, even though the full strategy for searching each database is present in Table 1 and Supporting information S2 of the current manuscript. This detailed full strategy information on the search terms and relevant parameters for each database, ensuring that the methodology is fully transparent and easily replicable.

Revised Location in Manuscript (file name: Revised Manuscript with Track Changes): Page 7, line 188-191

3. Comment: "3.In your PRISMA figure, please state the full precise exclusion reasons for the full-text reviewing stage."

Respond to comment 3:

We greatly appreciate your suggestion regarding the PRISMA flow diagram, which has further strengthened our protocol. We recognize the importance of providing clear and precise reasons for exclusions during the full-text review stage, as it enhances the transparency and rigor of the systematic review process.

Moreover, we have updated the PRISMA diagram to include the full and precise reasons for exclusions at the full-text review stage to address your concern. These reasons are clearly mentioned, which may improve the clarity and completeness of the manuscript.

Revised Location in Manuscript (file name: Revised Manuscript with Track Changes): Page 10, Figure 1 PRISMA flow diagram for systematic reviews

4. Comment: "4.First paragraph of the introduction is a bit too long. Consider summarising it."

Respond to comment 4:

We greatly appreciate your constructive comment. We agree with your point that the first paragraph of the introduction was somewhat lengthy and could benefit from greater conciseness to enhance readability.

To address your concern, we have revised the first paragraph, summarizing the key points more concisely while retaining the essential information. We believe the revision improves the readability of the introduction and ensures a clearer presentation of our objectives.

Revised Location in Manuscript (file name: Revised Manuscript with Track Changes): Page 3-4, line 61-79

---

## [Decision Letter · Decision Letter 2]

13 Jan 2025

PONE-D-24-15681R2

The roles and signalling pathways of lncMALAT1 in coronary artery disease: A protocol for systematic review of in vivo and in vitro studies

PLOS ONE

Dear Dr. Jufri,

Thank you for submitting your manuscript to PLOS ONE. After careful consideration, we have decided that your manuscript does not meet our criteria for publication and must therefore be rejected.

Specifically:

I am sorry that we cannot be more positive on this occasion, but hope that you appreciate the reasons for this decision.

Kind regards,

Academic Editor

PLOS ONE

Additional Editor Comments:

It seems to be a scoping review rather than a systematic review.

Reviewers' comments:

Reviewer's Responses to Questions

**Comments to the Author**

1. Does the manuscript provide a valid rationale for the proposed study, with clearly identified and justified research questions?

Reviewer #3: Yes

Reviewer #4: Yes

2. Is the protocol technically sound and planned in a manner that will lead to a meaningful outcome and allow testing the stated hypotheses?

Reviewer #3: Yes

Reviewer #4: Yes

3. Is the methodology feasible and described in sufficient detail to allow the work to be replicable?

Reviewer #3: Yes

Reviewer #4: Yes

4. Have the authors described where all data underlying the findings will be made available when the study is complete?

Reviewer #3: Yes

Reviewer #4: Yes

5. Is the manuscript presented in an intelligible fashion and written in standard English?

Reviewer #3: Yes

Reviewer #4: Yes

6. Review Comments to the Author

You may also provide optional suggestions and comments to authors that they might find helpful in planning their study.

Reviewer #3: The manuscript is suitable for publication in PLOS ONE as a protocol study, pending minor revision:

Consider adding a section elaborating on planned statistical methods for synthesizing and analyzing data.

Reviewer #4: Thanks to the authors for their accurate responses on my comments. I think the manuscript is now suitable to be accepted

7. PLOS authors have the option to publish the peer review history of their article (what does this mean? ). If published, this will include your full peer review and any attached files.

**Do you want your identity to be public for this peer review?** For information about this choice, including consent withdrawal, please see our Privacy Policy .

Reviewer #3: **Yes: ** Sareh Bakhshandeh Bavarsad

Reviewer #4: No

- - - - -

---

## [Author Response · Author response to Decision Letter 2]

12 Feb 2025

Respond to Reviewer #3 Sareh Bakhshandeh Bavarsad

Dear Prof. Sareh Bakhshandeh Bavarsad

Greetings.

We deeply appreciate your thorough contribution of our manuscript, and comprehensive review process, as well as the valuable comments provided. Your comments are valuable for further improving the manuscript. We have carefully considered the item and have added the following section of planned statistical methods for responses and revisions to address your concerns. Revised Location in Manuscript (file name: Revised Manuscript with Track Changes): Page 11, Line 276-286.

Respond to Reviewer #4

Dear Prof.,

Greetings.

We sincerely appreciate your time, effort, and thoughtful consideration in accepting our revised manuscript for publication. We are pleased to hear that our previous revisions have addressed your concerns. Moreover, we greatly appreciate the detailed and comprehensive review process, as well as the constructive feedback provided.

---

## [Decision Letter · Decision Letter 3]

24 Mar 2025

The roles and signalling pathways of lncMALAT1 in coronary artery disease: A protocol for systematic review of in vivo and in vitro studies

PONE-D-24-15681R3

Dear Dr. Jufri,

We’re pleased to inform you that your manuscript has been judged scientifically suitable for publication and will be formally accepted for publication once it meets all outstanding technical requirements.

Kind regards,

Ricardo Ney Oliveira Cobucci, Ph.D

Academic Editor

PLOS ONE

Additional Editor Comments (optional):

Reviewers' comments:

Reviewer's Responses to Questions

**Comments to the Author**

1. Does the manuscript provide a valid rationale for the proposed study, with clearly identified and justified research questions?

Reviewer #5: Yes

2. Is the protocol technically sound and planned in a manner that will lead to a meaningful outcome and allow testing the stated hypotheses?

Reviewer #5: Yes

3. Is the methodology feasible and described in sufficient detail to allow the work to be replicable?

Reviewer #5: Yes

4. Have the authors described where all data underlying the findings will be made available when the study is complete?

Reviewer #5: Yes

5. Is the manuscript presented in an intelligible fashion and written in standard English?

Reviewer #5: Yes

6. Review Comments to the Author

You may also provide optional suggestions and comments to authors that they might find helpful in planning their study.

Reviewer #5: I am happy to see that the authors have adequately revised the manuscript and addressed all the comments raised from the reviewers. I suggest publication.

7. PLOS authors have the option to publish the peer review history of their article (what does this mean? ). If published, this will include your full peer review and any attached files.

**Do you want your identity to be public for this peer review?** For information about this choice, including consent withdrawal, please see our Privacy Policy .

Reviewer #5: No

---

## [Editor Report · Acceptance letter]

PONE-D-24-15681R3

PLOS ONE

Dear Dr. Jufri,

I'm pleased to inform you that your manuscript has been deemed suitable for publication in PLOS ONE. Congratulations! Your manuscript is now being handed over to our production team.

Kind regards,

on behalf of

PROFESSOR Ricardo Ney Oliveira Cobucci

Academic Editor

PLOS ONE